# A User Study on Augmented Reality-Based Robot Learning Data Collection Interfaces

**Abstract:** Future versatile, generalist robots need the ability to learn new tasks and behaviors from demonstrations. Technologies such as Virtual and Augmented Reality (VR/AR) allow for immersive, visualized environments and settings that accelerate and facilitate the collection of high-quality demonstrations. However, it is so far unclear which interface is the most intuitive and effective for humans to create demonstrations in a virtualized environment. The intuitiveness and efficiency of these interface becomes particularly important when working with non-expert users and complex manipulation tasks. To this end, this work investigates five different interfaces in a comprehensive user study across various virtualized tasks. In addition, this work proposes a so far unexplored interaction interface, the combination of a physical robot for kinesthetic teaching with a virtual environment visualized through augmented reality. The environment, including all objects and a robot manipulator, is virtualized using an AR system. The virtual robot is controlled via various interfaces, i.e., Hand-Tracking, Virtual Kinesthetic Teaching, Gamepad, Motion Controller, Physical Kinesthetic Teaching. This study reveals valuable insights into the usability and effectiveness of these interaction interfaces. It shows that our newly proposed intuitive interface for AR control, i.e., using a physical robot as controller, significantly outperforms other interfaces in terms of success-rates and task completeness. Moreover, the results show that the motion controller and hand-tracking are also promising interfaces, in particular for cases where a physical robot is not available.

**Keywords:** Robot data collection, Human robot interaction

## 1 Introduction

Teaching robots new skills and tasks through demonstrations is an essential goal of the fields of robot learning and human robot interaction. The question how to learn tasks from demonstrations has received much attention through paradigms such as Imitation learning [1], Learning from Demonstrations [2] and Inverse Reinforcement Learning [3].

An important prerequisite for such approaches is the quality of the data and, hence, the data collection process itself. This requirement becomes even more important given the high data demand of recent learning methods. Prominent approaches focus on collecting demonstrations from various sources such as online videos [4] or dedicated first person videos [5]. However, demonstrating specific tasks to a particular platform harbours several challenges for real world experiments, e.g., reproducibility issues due to changing objects and object poses, cumbersome and slow resetting of experimental setups and inaccurate measurements due to sensor noise. Virtualisation alleviates many of these challenges as it allows for highly reproducible and controllable experiments that can be quickly reset and repeated. While some approaches utilize screens to virtualize experiments [6], leveraging augmented or virtual reality (AR/VR) offers significant advantages [7] , by providing an immersive experience that allows intuitive control over the perspective onto the virtual environment.

Submitted to the 7th Conference on Robot Learning (CoRL 2023). Do not distribute.

While research has highlighted the advantages of AR/VR headsets over screens for visualization purposes[7], there has not yet been a study investigating the comparative performance of different interaction methods within AR/VR environments for collecting task demonstrations for robots. This paper closes this gap by presenting a comprehensive study on different ways to interact with virtual experiments for the sake of collecting demonstrations in an AR/VR setting. This study compares five different options to interact with the virtual environment, i.e., inside-out Hand Tracking, Kinesthetic Teaching of a virtual robot, end-effector control via gamepad, end-effector control via VR Motion Controller and Kinesthetic Teaching via a physical robot. A total of 35 participants took part in the study, whereas each participant utilized every interface to collect up to 3 demonstrations across 3 tasks with varying difficulty levels. The performance of the interfaces were evaluated across several dimensions including objective measures, such as success rate and completion time of the demonstrated tasks as well as subjective measures surveyed via the well established modular extension of the User Experience Questionnaire (UEQ+) [8].

Our study shows that combining physical Kinesthetic Teaching with AR provides a powerful and intuitive system to efficiently collect demonstrations in virtual environments. The study further reveals that this system significantly outperforms any other interface with respect to both the objective and subjective measures resulting in an effective and intuitive system that allows for efficient and reliable data collection in virtual environments. Such an interface could also be easily implemented for controlling a real physical robot via tele-operation, yet, this is part of future work.

In summary, the contributions of this paper are twofold. First, a comprehensive study of different interaction interfaces for the purpose of collecting virtual demonstrations in an AR/VR setting. Second, introducing a new interaction interfaces that utilizes a physical robot platform for controlling a (virtual) robot in a virtual experimentation setting.

## 2 Background

**Robot Interface** Demonstrations can be broadly categorized into three categories, Kinesthetic Teaching, teleoperation and pure observation. Given, the goal of efficiently collecting demonstrations in a virtual environment, we identified five promising interaction interface in the literature. **Hand Tracking** uses sensors, usually cameras, to track the hand of the user and maps the robot state to the hand used in teleoperation [9, 10] and shared-control telemanipulation [11, 12] scenarios. **Virtual Kinesthetic Teaching** interfaces allow the manipulation of a virtual robot directly using the participants hands, for example to teleoperate physical robots in bi-manual [13] and digital twin [14] settings. **Gamepads** have been widely used to control physical robots while adding very little system complexity, specifically in the area of teleoperation [15, 16]. Recently **Motion Controllers** have become increasingly popular in both robot learning and teleoperation[17], especially because of haptic and AR visual cues [18]. **Kinesthetic Teaching** commonly refers to the manipulation of a physical robot for the purpose of collecting demonstrations directly on that platform[19]. However, it also provides a very intuitive and straight forward demonstration interface for teleoperation systems[2]. While there has been some preliminary work investigating physical Kinesthetic Teaching controlling a virtual twin [20], it does not leverage the advantages of combining this interface with a AR/VR system.

**AR & VR for Robotics** The ability to directly render and immerse users in a 3D virtual environment makes AR/VR technologies a promising tool for collecting task demonstrations. In combination with haptic feedback, these technologies have been shown to provide higher teaching efficiency than common GUIs [7]. It can reduce workload [21], enhance the accuracy and speed of collaborating users [22] and improve overall task performance, compared to non-AR baselines [23].

Other approaches use virtual reality to realize teleoperation. including robot arm [24, 25], mobile robot [25], bimanual robot arm [26, 27, 14], humanoid robots [28, 29], and surgery robots [30]. Apart from VR-based method,

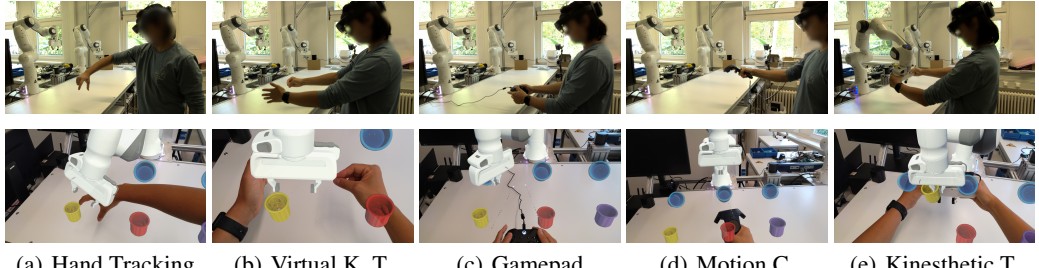

|(a) Hand Tracking|(b) Virtual K. T.|(c) Gamepad|(d) Motion C.|(e) Kinesthetic T.|

Figure 1: The top row shows a participant collecting demonstrations using the different interfaces from an outside viewpoint. The bottom row shows the virtualized environment as it is presented to the participant via the HoloLens 2.

## 3   Technical Details

This study explores diverse interaction interfaces in the context of their effectiveness and intuitiveness for gathering demonstrations within AR environments. To achieve this, a novel framework has been developed, that seamlessly integrates a game engine responsible for rendering the virtual environment on AR headsets with a physics simulator, enabling the simulation of both the virtual robot and manipulated objects.

**Physics Simulator**   The framework deploys the widely applied and utilized MuJoCo physics simulator[31]. The virtual environments, including various manipulatable objects and a Franka Emika Panda Robot were implemented within the simulator and combined into several meticulously designed scenarios. Additional data loggers were implemented that record state information of the virtual robot as well as the virtual objects, including position, velocity, acceleration and orientation.

**Augmented Reality Platform**   The virtual scene is presented to the user via the Microsoft HoloLens 2 [32], notable for its diagonal field of view (FOV) measuring 96.1 degrees. Leveraging the Unity Engine, a custom AR application specifically tailored for use with the HoloLens 2 was developed. The HoloLens 2 renders all virtual elements, including the robot and the objects, in real-time, providing users with an immersive and interactive experience.

**Interaction Interfaces**   This study investigates five different interaction interfaces that have increasing hardware demands beyond an AR/VR Headset. For every interface but Kinesthetic Teaching a IK solver was used to determine the robot configuration, given the end effector pose. The Inside-out **Hand Tracking** (shown in Figure 1(a)) interface uses two scene cameras of the AR Headset to track the hand and recognize different gestures. To increase the level of intuitiveness and immersion, the gripper of the virtual robot is aligned with the index finger and thumb of the tracked hand. The participants can intuitively control the robot's movements by moving their hand around while closing and opening the gripper by executing a pinch or release motion with their index finger and thumb. Similar to the Hand Tracking interface the **Virtual Kinesthetic Teaching** (shown in Figure 1(b)) also allows for the direct control of the virtual robot without additional hardware. The participants can move the robot by grabbing the virtual end effector. Releasing the end effector stops the tracking. Stretching and squeezing gestures trigger the virtual robot to close and open its gripper respectively. The **Gamepad** interface (shown in Figure 1(c)) uses a Microsoft Xbox controller to manipulate the virtual robot. The **Motion Controller** interface (shown in Figure 1(d)) uses a Vive Pro Motion Controller 2.0, which precisely measure the Motion Controller's position and orientation in real-time. The end effector pose of the virtual robot is mapped to the pose of the Motion Controller and the virtual gripper is opened and closed by holding and releasing the triggers of the controller. The **Kinesthetic Teaching** interface (shown in Figure 1(e)) allows participants to directly control a physical version of the virtual robot. The physical robot transmits joint positions and velocities to the virtualization framework in real-time, mapping the configuration of the virtual

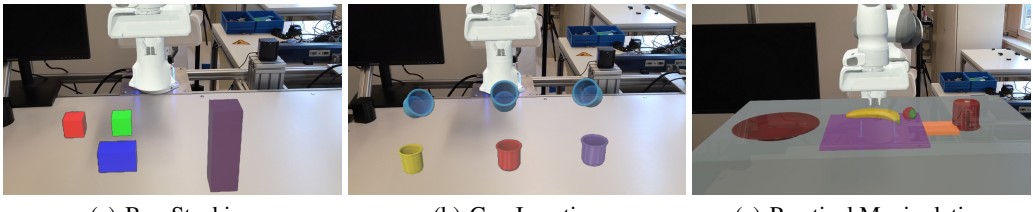

|           (a) Box Stacking           |           (b) Cup Inserting           |     (c) Practical Manipulation     |

Figure 2: Tasks used in the user study. (a) Box Stacking task: Stack three boxes within the purple area. (b) Cub Inserting task: Insert three cups with varying diameters into larger cups. (c) Practical Manipulation task: Move fruits onto the plate, move the plate into the purple area and flip the mug and position it on the orange area.

robot to the physical one. While this interface provides the most detailed control of the virtual robot it also has the limitation that access to the actual physical robot is required.

## 4   User Study

In order to assess the efficiency and intuitiveness of various interfaces for collecting demonstrations, we designed a comprehensive user study. This study aims to evaluate each interface thoroughly and establish meaningful comparisons among them.

**Questionnaire**   The user study consisted of two questionnaires (background and interface assessment) for each participant. The first questionnaire (background) includes seven questions and aims at the theoretical and practical knowledge of the participant in terms of using physical robots, AR/VR/MR devices, and the Gamepad. The answers were given as multiple choice in an explicit way, to avoid subjective scale measuring [33] and misunderstandings.

The second questionnaire (interface assessment) measures the subjective assessment from participants with regard to the usage of the five different interfaces. The questionnaire itself consists of five scales taken from UEQ+ [8], including **attractiveness**, **efficiency**, **perspicuity**, **dependability** and **novelty**. Each scale presents four pairs of contrastive adjectives along with a scale ranging from one to seven, where four is neutral.

**Study Procedure**   The user study starts with participants filling out the background questionnaire. Afterwards, each participant was randomly assigned to one task and provided with a corresponding video to understand the task objectives. The order in which participants use the five interfaces was randomized, to prevent potential biases. Before each interface usage, the participants had one minute to get familiar with the corresponding interface. Subsequently, participants used each interface three times, resulting in a potential learning curve over the three demonstrations. Each demonstration must be completed within a specific time frame, otherwise the current demonstration was stopped and the next one started. After the completion of three demonstrations with one interface, participants were asked to fill out the interface assessment questionnaire, to indicate their impressions and experiences in regards to the corresponding interface.

**Metrics**   The study contains objective and subjective metrics. For objective metrics, we look at task success (did the participant finish the whole task?), task completeness (how many sub-goals did the participant fulfill?), and required task completion time (how much time did the participant need to finish the task?). The subjective metrics are based on the interface assessment questionnaire.

**Study Tasks Design**   The user study included three different tasks, box stacking, cup stacking and practical manipulation. The **box stacking task** serves to assess the basic pick and place skills of the interface. Participants were asked to place and stack three boxes in 60 seconds into a target area, as you can see in Figure 2(a). For each box stacked successfully, this demonstration will give 0.3

Table 1: The number of demonstrations per task and interface

|                              | Task 1 | Task 2 | Task 3 |
|------------------------------|--------|--------|--------|
| Gamepad                      | 30     | 36     | 29     |
| Hand Tracking                | 31     | 40     | 27     |
| Kinesthetic Teaching         | 35     | 37     | 32     |
| Motion Controller            | 31     | 32     | 30     |
| Virtual Kinesthetic Teaching | 30     | 35     | 28     |

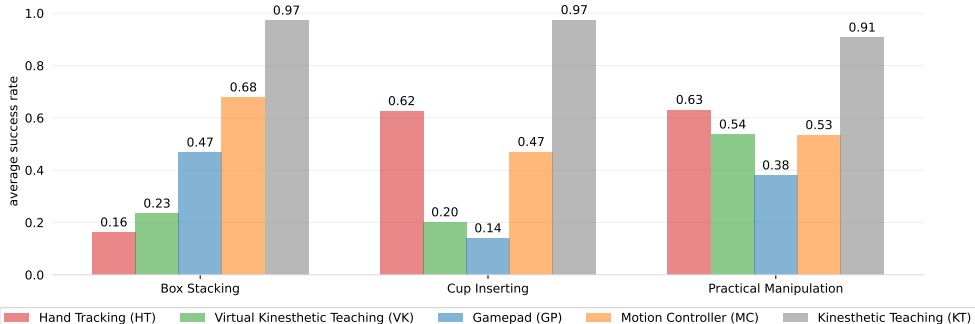

Figure 3: This graph shows the success rates of each interface across the different tasks averaged over all task executions. A task execution is considered successful if all sub-tasks were fulfilled. **Kinesthetic Teaching** consistently maintains the highest success rate of over 0.90 in all three tasks. The poor performance of the **Gamepad** interface in task 2 indicates that this interface is a poor fit for tasks that require precise control of the orientation of the end-effector. Hand tracking reaches the second-highest success rate in tasks 2 and 3, but it exhibits the lowest success rate in task 1. The Motion Controller has a success rate ranging from 0.47 to 0.68 across all three tasks. Virtual Kinesthetic Teaching performs a success rate of approximately 0.2 in tasks 1 and 2, while it achieves a higher success rate of over 0.5 in task 3. In tasks 1 and 3, the Gamepad interface yields success rates of 0.47 and 0.38, respectively. However, it only attains a success rate of 0.14 in task 2.

completeness score and a final 0.4 for the last box. The **cup stacking task** is designed to evaluate the flexibility and precision of the interfaces to do dexterous motion. Participants were asked to insert three different sized cups in 60 seconds into three corresponding tilted cups, as seen in Figure 2(b). The completeness gain per cup is 0.25, 0.35, and 0.4. The **practical manipulation task** is designed to evaluate the comprehensive manipulation ability of each interface in a longer sub-task sequence and is limited to 90 seconds. It involves five steps (shown in Figure 2(c)), including placing a banana on a plate, placing a strawberry next to the banana, pushing the plate into a target area, flipping a mug, and placing it on a specific location. Each successful step will gain a completeness of 0.2.

**Participants**   The user study included 35 participants from the local and other universities, containing six females and 29 males, aged between 15 and 30. Each participant used all five interfaces three times for the random assigned task. Some demonstrations had to be discarded due to system instability, failed records or hardware issues. Finally, we collected 483 valid human demonstrations which were executed on the different interfaces.

## 5   Result

**Success Rate**   As depicted in Figure 3, Kinesthetic Teaching has the highest success rate with a large margin over the other methods, with a success rate of above 90% across all tasks. Given that the success rate follows a binomial distribution with ties between the interfaces for each participant, we use a **Mann-Whitney U test** [34] to analyze the significance of the differences in the success rates. To avoid dependencies across demonstrations of the same participant, we only used the last of the three demonstrations created by each participant. All following Mann-Whitney U tests are

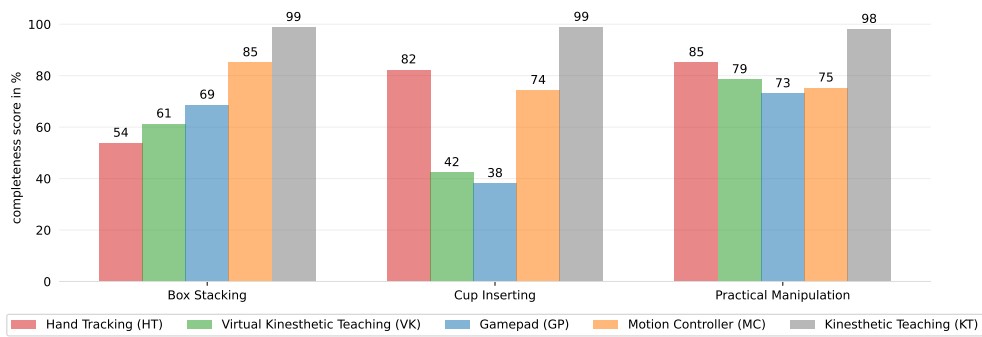

Figure 4: This graph shows the task completeness for each interface across the different tasks averaged over all task executions. The completeness of an execution is based on how many sub-tasks were achieved.

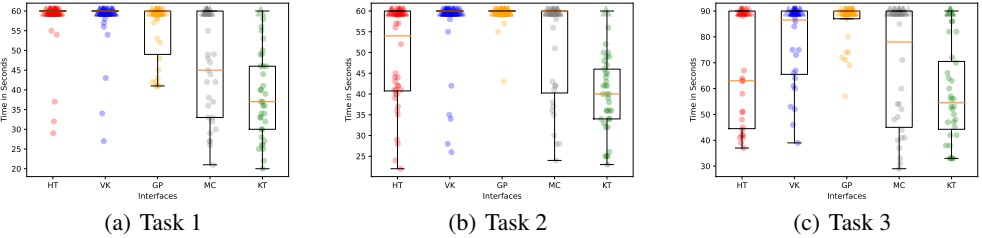

(a) Task 1        (b) Task 2        (c) Task 3

Figure 5: This graph shows the time spent on each demonstration by participants with each interface in each task. The dots and triangles represent successful and unsuccessful demonstrations. The yellow line in each box shows the mean time of every demonstration, and the time of each failed demonstration is counted as the maximum time limitation.

computed analogously. In all 3 tasks, the Mann-Whitney U test revealed that **Kinesthetic Teaching** significantly outperformed the other four interfaces (excluding Motion Controller from task 1 and Hand Tracking from task 2). Regarding the other interfaces, one key finding is that the **Motion Controller** interface performed reasonable on all tasks, outperforming Virtual Kinesthetic Teaching and Gamepad on most tasks. **Hand Tracking** performed well on task 2 and 3, even with a margin over the Motion Controller, but under performed on the first task that required picking the blue box and reorienting it, which caused problems with the internal Hand Tracking of the Holo-lens 2. The **Gamepad** and **Virtual Kinesthetic Teaching** showed the worst performance, in particular for tasks that require a precise control of the orientation of the end-effector (cup inserting).

**Task Completeness** For task completeness, the results are shown in Figure 4. They again confirm our finding that **Kinesthetic Teaching** outperforms all other interfaces. The Kinesthetic Teaching consistently provides a very high completeness of 98%. A Mann-Whitney U test shows the same significant difference, as in the success metric, to all other interfaces (again excluding the Motion Controller from task 1 and the Hand Tracking from task 2).

**Task Completion Time** The mean completion time for the different interfaces is shown in Figure 5. The **Kinesthetic Teaching** interface allowed for the fasted task completion times as most trials could be completed with this interface while other interfaces had a worse success rate. We again performed a Mann-Whitney U test to confirm the significance of the results. The test confirmed that the **Kinesthetic Teaching** allows for faster task execution than the Gamepad ($p < 0.01$) and Hand Tracking ($p < 0.001$) in task 1 and task 2, Virtual Kinesthetic Teaching ($p < 0.005$) in all 3 tasks. Moreover, the **Motion Controller** has a significant difference with Gamepad (0.05) in task 1 and task 2, **Hand Tracking** ($p < 0.001$) in task 1.

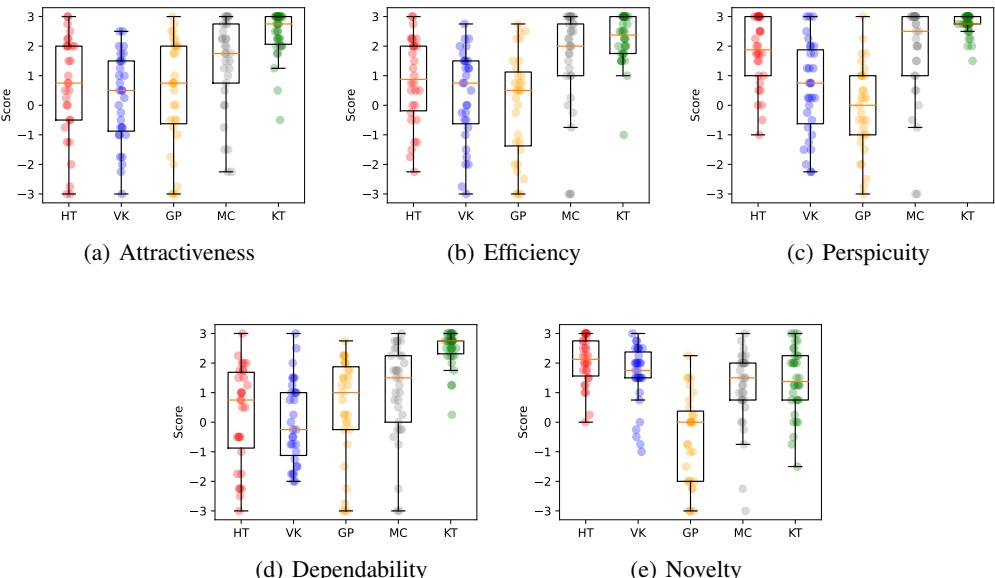

(a) Attractiveness

(b) Efficiency

(c) Perspicuity

(d) Dependability

(e) Novelty

Figure 6: This graph shows the subjective metrics taken from the assessment questionnaire for every interface. including attractiveness, efficiency, perspicuity, dependability, and novelty. **Kinesthetic Teaching** emerged as a standout performer, boasting a significant score with a relatively lower variance over the other four interfaces, except for Novelty. It means that most of the participants are really satisfied with this interface.

**Subjective Metrics**   The subjective metrics are shown in Figure 6. Our study revealed the following insights: (i) **Kinesthetic Teaching** emerged as a standout performer, with a higher score with relatively small variance over the other four interfaces, except for Novelty, where it was on par with the other methods. This pattern was consistent in the categories attractiveness and dependability, with Kinesthetic Teaching exhibiting a noteworthy difference when compared to any of the other four interfaces, while no significant differences were observed among the latter. The efficiency scale unveiled a similar trend, however, here also the **Motion Controller** interface showed a significant performance difference from the Gamepad, indicating its superior efficiency in comparison to this particular interface. In terms of Perspicuity, we identified significant differences between **Motion Controller** and **Hand Tracking** when contrasted with **Virtual Kinesthetic Teaching** and **Gamepad**, highlighting variations in their ease of understanding and clarity. Lastly, for Novelty, **Hand Tracking** stood out as the most appealing interface, underlined by its significant difference from Motion Controller and Gamepad. Conversely, **Gamepad**, emerged as the least novel interface, significantly differing from all other interfaces in this regard. These subjective metrics offer valuable insights into the user perception and preferences associated with each interface, providing a holistic understanding of their strengths and weaknesses.

**Background Analysis**   We further analyzed the impact of user experience with the Gamepad, and its high-frequency use, on the performance of the participants. Interestingly, experience with the Gamepad positively influenced performance across all interfaces, not limited to the Gamepad interface alone. This suggests that the skills and familiarity gained from using a Gamepad (typically connected with playing computer games) can be transferable and advantageous when navigating various interfaces. Furthermore, we explored whether there were differences in the success rate collected through Kinesthetic Teaching based on participants regular physical activity. Here, our analysis showed no significant difference in the success rates. This result implies that the effectiveness of Kinesthetic Teaching remains consistent regardless of participants physical activity levels, highlighting the accessibility and inclusivity of this teaching approach.

**Further Discussion** **Hand Tracking** exhibits substantial potential as an interface for robot inter- action. It stands out by delivering performance that competes favorably with the Gamepad interface. Moreover, it offers the promise of generating high-quality demonstrations and presents a relatively gentle learning curve for users. In our user study, we observed a notable phenomenon: users were more likely to encounter robot singularity when employing Hand Tracking. This observation might be attributed to differences in the robot's configuration and arm design compared to other interfaces. Further exploration of these factors is warranted to better understand the specific challenges and advantages associated with Hand Tracking in robot interaction contexts.

**Virtual Kinesthetic Teaching** represents an innovative approach to robot interaction; however, its performance lags behind that of the Gamepad interface. The participants highlighted a significant limitation: they were unable to perceive the presence of the robot during Virtual Kinesthetic Teach- ing sessions, which contributed to its underwhelming performance. Additionally, the stability of Hand Tracking and gesture recognition emerged as a critical factor influencing the overall user ex- perience with Virtual Kinesthetic Teaching. These technical aspects greatly impacted the interfaces effectiveness and user satisfaction.

The **Gamepad** interface presents an economical option for generating human demonstrations. Nev- ertheless, it does come with certain drawbacks, including inefficiency, a steep learning curve, and extensive user experience. Dexterous manipulation, e.g., the positioning of one cup inside another with a particular orientation, as well as complex 3D trajectories requried of objects as well as a lim- itation of this interface. Despite these challenges, the Gamepad interface does exhibit a respectable success rate in both Task 1 and Task 3, exceeding 68%. This suggests its capability to handle funda- mental tasks such as object manipulation—particularly picking and placing. Those with more exten- sive experience are more likely to produce high-quality human demonstrations using this interface. Furthermore, it is worth noting that the Gamepad interface is sensitive to users prior experience.

**Motion Controller** interface has proven to be both efficient and user-friendly. Participants partic- ularly appreciated the simplicity of gripping objects by merely holding the trigger of the Motion Controller. This feature was deemed more convenient in comparison to Kinesthetic Teaching. The Motion Controller interface also exhibited commendable completeness across all three tasks, with a success rate exceeding 74%. Users also noted that they could provide demonstrations in a shorter amount of time using this interface.

**Kinesthetic Teaching** emerged as the most potent interface in our user study. It outperformed other interfaces with its high-precision motion control and a consistently high success rate. However, the main drawback was that users were required to have access to a physical robot to create human demonstrations, which may not be feasible for all potential users. Additionally, some participants encountered difficulties, noting that the end effector of the physical robot was heavy to manipulate and that closing the grippers required more effort.

## 6 Conclusion

In this paper, we conducted a comprehensive study with 35 participants of different interaction inter- faces for collecting virtual demonstrations in AR setting. From the result, the Kinesthetic Teaching with AR has a highlighted performance over almost all the metrics. Due to its limitation of complex hardware setting, we also conclude that Motion Controller and Hand Tracking is also a promising way to create high-quality human demonstration. For virtual Kinesthetic Teaching, it was considered as a novelty way but it didn't have a good performance during this study. Gamepad, as a low-cost way to create human demonstration, has worked well in simple tasks such as pick and place, but it is not feasible for complex 3D manipulation. Our research will help the further studies about learning from demonstration, and may provide them with a novel perspective to create and study how human demonstrations influence the intelligent agent.

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
