# OpenReview forum: "A User Study on Augmented Reality-Based Robot Learning Data Collection Interfaces"
_robot-learning.org/CoRL/2023/Workshop/TGR — CoRL 2023 Workshop TGR Poster_

### Official Review · Reviewer_axbw · 2023-10-19
**Strong accept**

**Rating:** 8
**Confidence:** 4

**Review:**

Very nice study over AR-based robot learning data collection interfaces. Glad to see the comparison result between multiple interfaces, especially over several subjective metrics. Clearly a good contribution to the community.

---

### Decision · Program_Chairs · 2023-10-20

**Decision:**

Accept (Poster)

**Comment:**

Great paper and closely aligned topic!